# Impact of Midurethral Sling Implantation on Sexual Function in Women with Stress Urinary Incontinence

**DOI:** 10.3390/jcm9051538

**Published:** 2020-05-20

**Authors:** Edyta Horosz, Aneta Zwierzchowska, Andrzej Pomian, Wojciech Majkusiak, Paweł Tomasik, Ewa Barcz

**Affiliations:** Multidisciplinary Hospital Warsaw-Miedzylesie, Department of Obstetrics and Gynecology, 04-749 Warsaw, Poland; edytahorosz@tlen.pl (E.H.); a.j.zwierzchowska@gmail.com (A.Z.); apomian@gmail.com (A.P.); wmajkusiak@gmail.com (W.M.); p_tomasik@wp.pl (P.T.)

**Keywords:** midurethral sling, urinary incontinence, female sexual function, quality of life

## Abstract

Stress urinary incontinence (SUI) negatively influences sexual functions. However, the available data on sexual activity of patients who underwent midurethral sling (MUS) implantation are inconsistent. Our aim was to evaluate the impact of MUS implantation on sexual functions of women with SUI. We enrolled 171 patients undergoing the MUS procedure. Preoperative examination included the cough test, 1 h pad test and the Prolapse/Urinary Incontinence Sexual Questionnaire, IUGA Revised (PISQ-IR). All patients had the retropubic sling implanted. Follow-up visits were performed 6–12 months after surgery. Objective cure rate was obtained in 90.98% of patients. Coital incontinence was reported by 56% of women before the surgery, and 8.6% afterwards. Among women who gained continence, significant improvement in sexual function was observed in the majority of the domains. In women who were not objectively cured (9.02%), we did not observe improvement in sexual life. All these patients indicated fear of leaking urine during sexual activity as the main cause of avoiding sex, similarly as before operation. To conclude, successful treatment of SUI with MUS significantly improves the quality of sexual life. On the other hand, persistent incontinence appears to be the most probable cause of lack of improvement in the quality of sexual life.

## 1. Introduction

Stress urinary incontinence (SUI) is a health issue that affects social, physical, psychological, domestic, occupational as well as sexual well-being [1]. Female sexuality is complex and multifactorial. It appears to be influenced by organic and anatomic conditions, as well as psychological factors. Studies found that women with urinary incontinence (UI) report lower intercourse frequency, avoidance and even complete abstinence of sex [2]. Coital incontinence in women seeking therapy for UI was reported to range from 10% to 56% [3]. Fear of UI during sexual intercourse, decreased libido, irritation of the vulvovestibular region from persistent urine leakage, deficits in lubrication and dyspareunia are common sexual complaints. Accompanied by impaired body image, reduced self-esteem and avoidance of sex, those symptoms lead to female sexual dysfunction (FSD). It is estimated that FSD may be experienced by more than 60% of women suffering from UI [4]. A comparably high prevalence of sexual problems in women suffering from UI is reported by patients suffering from vulvovaginal atrophy associated with estrogen deficiency [5].

Having acknowledged that the physical aspect of UI is one of the important factors responsible for sexual dysfunction, it is reasonable to assume that the resolution of incontinence will positively influence the quality of sexual life. However, the available data on sexual activity and function of patients who underwent midurethral sling (MUS) implantation are inconsistent. Some authors report improvement [6,7], whereas others show deterioration [8,9]. There also exist data that are equivocal [10,11]. A meta-analysis including 21 studies revealed no difference in sexual function in more than half of all women (55.5%) who underwent different types of incontinence surgery [12]. It may be associated with failure of the procedure, alteration of vaginal anatomy, elevation or narrowing of vaginal wall, reduced sensation, dyspareunia, de novo urgency occurring after the surgery, but also with psychological and emotional factors [13].

The aim of the present study was to evaluate the impact of MUS implantation on sexual activity and function in women with SUI. We also analyzed sexual function in cases of failed SUI treatment, in order to determine whether urine leakage itself may be the cause of lack of improvement.

## 2. Materials and Methods

The present study was performed during the years 2018–2020 as a prospective cohort study. We enrolled 171 consecutive patients undergoing the MUS procedure. The inclusion criteria comprised SUI and consent for surgery.

In all cases, SUI was confirmed with medical history and cough stress test in semi-sitting position with bladder filling ca. 300 mL. The bladder filling was evaluated with the use of ultrasound examination (transabdominal probe). Preoperative examination included 1 h pad test, urogynecological physical examination and self-administered questionnaires. Incontinence Impact Questionnaire 7 (IIQ−7) was used to assess the impact of UI on health-related quality of life. Sexual function was assessed with the Prolapse/Urinary Incontinence Sexual Questionnaire, IUGA Revised (PISQ-IR)—a condition-specific, validated and reliable instrument that evaluates sexual function in women with UI and/or pelvic organ prolapse (POP), and measures physical, behavioral-emotive and partner-related domains [14]. PISQ-IR questionnaire consists of two parts. Part 1, for non-sexually active (NSA) women, includes 12 items in the following scales: Condition Specific (NSA-CS, 3 items), Partner-related (NSA-PR, 2 items), Global Quality (NSA-GQ, 4 items), and Condition Impact (NSA-CI, 3 items). Higher scores indicate greater impact of the condition on sexual inactivity. Part 2, administrated to sexually active (SA) women, includes 21 items in the following scales: arousal, orgasm (SA-AO, 4 items), Partner-related (SA-PR, 3 items), condition specific (SA-CS, 3 items), global quality (SAGQ, 4 items), condition impact (SA-CI, 4 items), and desire (SA-D, 3 items). Higher scores indicate better sexual function [15].

The exclusion criteria included history of previous anti-incontinence surgery and concomitant POP equal to or greater than degree II, assessed using the POP quantification (POPQ). All patients had the retropubic sling implanted (Gynecare TVT blue, Ethicon Inc., Johnson & Johnson, Somerville, NJ, USA). The surgical procedure was performed under general anesthesia, according to the 1/3 rule, after pelvic floor ultrasound examination of urethral length (the location of the distal end of the vaginal incision from the external urethral orifice equaled one third of the sonographic urethral length) [16]. In all cases, tensioning of the tape was achieved with the use of intraoperative cough test.

Follow-up visits were scheduled 9–12 months post-surgery (mean 10.4 months; some of the patients were examined earlier than 9 months, for their convenience), and included cough test, 1 h pad test and pelvic floor ultrasound examination. All patients completed the IIQ−7 and PISQ-IR. Objective cure rate was defined as no leakage during cough test with bladder filling ca. 300 mL and negative 1 h pad test (≤2 g). We also performed analysis of intra and postoperative complications such as urinary retention (post-voiding residual volume > 100 mL), de novo overactive bladder syndrome, vaginal mucosal erosion, pain and dyspareunia

The primary objective of the study was to evaluate the influence of the successful TVT (tension-fee vaginal tape) procedure on different aspects of sexual life. The second objective was to analyze various aspects of sexual life in cases in whom failure of the anti-incontinence procedure was diagnosed.

Descriptive statistical analysis and statistical tests were performed using the R version 3.4.0 (by the R Foundation for Statistical Computing, Vienna, Austria). U Mann Whitney, paired T-tests and Wilcoxon signed-rank tests were used to compare quantitative variables. For categorical data, the chi-square test was used. A *p*-value < 0.05 was considered significant.

The study was approved by the local Medical Ethical Committee (KB/1264).

## 3. Results

Out of the 171 patients enrolled, 116 (67.8%) were sexually active (sexual intercourse within 6 months before surgery, and intention to be active after the surgery) and 55 (32.2%) were not (no sexual intercourse within 6 months and no intention to initiate sexual activity after the surgery). Of the 133 women who accomplished the 6–12 months follow-up, 86 (64.6%) were sexually active, 47 (35.4%) were not. The mean age of the patients was 56.3 ± 11.2 years. The mean BMI was 27.8 ± 4.54. 94 (55%) patients were postmenopausal. The only demographic differences found in the sexually active and non-active groups were age, body weight and BMI (Table 1). 34 patients (20%) were lost to follow up. In 24 cases, the questionnaires were incomplete (exclusion criterium). In 5 cases, patients did not attend the follow-up visit, three women had been sexually active before surgery but ceased to be active afterwards, whereas two women had been non-active and started having intercourse after the procedure. These 5 patients were also excluded from the study.

We obtained objective cure rate (1 h pad test ≤ 2 g) in 90.98% of patients. Statistically significant improvement was also observed in the quality of life assessed with the IIQ7 (Incontinence Impact Questionnaire-7) (Table 2). The rate of coital incontinence was 56% before the surgery and 8.6% after the procedure. No severe bleeding, bladder perforation or other intraoperative complications occurred. In four cases, tapes were removed within seven days from the operation because of voiding difficulties and urine retention after micturition (retention > 100 mL, voiding in semi-sitting position, pain, intermittent urine flow), and inappropriate sling location determined with ultrasound (sling located < 3 mm from the urethra). During the follow-up evaluation we observed no vaginal erosion. Of the 12 women who experienced surgical failure, 4 reported urgency and 3 of them had urinary retention. The retention (mean 56.4 mL, accompanied by recurrent bladder infection or urgency as a complication) in those three patients was diagnosed during the follow-up visit. The sling location was correct and no retention was observed on discharge day.

The proportion of women who were sexually active was similar before and 6–12 months after the anti-incontinence surgery (67.8% vs. 64.6%, *p* = 0.76). Among women who were continent after the procedure, significant improvement in sexual function was observed in the majority of the domains. We observed global improvement in the quality of sexual life. Obtaining continence reduced fear and shame during sexual activity (condition specific, CS), as well as feeling sexually inferior, embarrassed or angry during sexual activity (condition impact, CI). Moreover, gaining continence positively influenced frequency of sexual activity (CI), improved contentment, satisfaction and sexual self-esteem. There were no differences in the level of sexual interest and sexual desire before and after surgery, but there was a statistically significant improvement in sexual arousal during intercourse. No changes were also observed in the partner related domain (Table 3).

During the 6–12 months follow-up, 12 patients (9.02%) had a positive pad test result. Eight of them reported persistent SUI, four had mixed UI. In this group we did not observe improvement in sexual life. Incontinence during sexual activity concerned each woman in this group, and all patients indicated fear of leaking urine during sexual activity as the main cause of avoiding sex, similarly as before operation. This phenomenon was not correlated with the amount of urine leak observed during the pad test. Surgery did not change the intensity of dyspareunia, but no cases of de novodyspareunia were observed. Three of the eight sexually active women indicated urgency as the second cause of dissatisfaction and feeling embarrassed about sexual life. There was no change in the partner-related domain, or the level of sexual desire or interest (Table 4).

## 4. Discussion

According to the World Health Organization (WHO), quality of life is defined as the individual’s perception of their position in life, in the context of the culture and value systems in which they live, and in relation to their goals, expectations, standards and concerns. Sexual life is one of the important factors that plays a role in the feeling of health and good quality of life. It depends on many interrelated elements, including physical, emotional, mental, social and biological factors [17]. Even one disturbed element may influence the whole sexual self-esteem, and may lead to avoidance or abandonment of sexual activity [18].

SUI and POP have a negative impact on female sexual functioning, and surgical treatment can be effective in improving the quality of global and sexual life [2,19]. SUI is predominant in young and middle-aged women who are more likely to be sexually active [20]. Therefore, this condition may have a serious impact on the quality of sexual life in these women. Coital incontinence that occurs mainly at penetration is a sexual symptom that appears to be very common and troublesome. The prevalence of urinary leakage during sexual intercourse ranges from 10% to 56% among incontinent women [3].

It has been shown that midurethral slings are the most effective treatment for SUI, with objective cure rates ranging from 70% to even over 90% [21]. We therefore assumed that surgery may be beneficial for these women. However, data concerning female sexual function after correction of incontinence with slings appear contradictory and controversial, mainly because of a number of limitations to those studies, such as retrospective design, heterogeneity of patient samples with respect to the type of UI, coexisting POP, previous or concomitant surgeries, and different methods used to asses, analyze and report sexual function.

In the current study we obtained statistically significant improvement of sexual quality of life in the analyzed group. Our results are in contrast with a study conducted by Mazouni et al. that showed no change in sexual life in 74.4% of women who underwent the TVT procedure. Improvement was observed only in one patient, and deterioration in 20% of women who complained of dyspareunia (14.5%) and loss of libido (5.4%). It is worth mentioning, however, that despite the high cure rate (87.3%), the authors noticed postoperative voiding difficulties, such as incomplete voiding, double voiding and straining to void in 60% of women. 47.2% of the patients complained of urgency and 32.7% reported frequency. The authors did not analyze separately the cured patients without any complications following the TVT procedure and women who suffered from voiding difficulties. It is plausible that these complications influenced sexual life and the results of the questionnaire [8]. Similarly, Maaita et al. reported no change in sexual function in 72% of women after implantation of the TVT. The patients (14%) that reported deterioration indicated loss of libido as the main cause, but none of the women complained of dyspareunia, any loss of sensation or achievement of arousal, or tenderness over the scar in the anterior of vaginal wall. Most of the patients (63%) had undergone previous abdominal or vaginal surgery, which might influence quality of sexual life. Moreover, the results of this study are based solely on a questionnaire completed by the patients 6–36 months after the surgery. No clinical examination was performed during the follow-up, hence it is difficult to assess whether there were other factors that could affect the results [22].

In the current analysis we observed improvement in the quality of sexual life. However, it must be stressed that we obtained continence and avoided complications in over 90% of cases. A separate analysis of cured and failed cases (persistent SUI and complicated cases) showed that deterioration was associated with persistent SUI, OAB (overactive bladder), urgency and fear of urine leakage during intercourse. Therefore, treatment failure or postoperative complications appeared to constitute one of the most important factors responsible for the impairment of sexual functioning. In previously cited studies, the authors did not analyze separately successful and failed cases. It is likely that this fact contributes to the differences in the results when compared to ours.

We postulate that the positive impact of TVT on sexual functioning is primarily attributable to the resolution of incontinence after surgery. We observed overall improvement of the quality of sexual life, mainly due to the absence of incontinence during sexual activity and absence of fear of incontinence. This is in keeping with Ghezzi et al., who demonstrated significant improvement in the sexual functioning of all women who reported resolution of UI after the TVT procedure. The authors found no significant differences in the incidence of dyspareunia, and only 2 out of 53 patients reported deterioration in the quality of sexual life after the surgery. In one case, this was due to vaginal exposition of the tape, whereas in the other one, de novo anorgasmia was stated as the main cause [23]. Our findings, obtained in a larger patient sample, support the conclusions of the cited authors.

The results of the current analysis also imply that complications of the surgery, such as de novo OAB, voiding difficulties or persistent incontinence, impact sexual functioning negatively. This is inagreement with a study conducted by Elzevier et al., who evaluated sexual function after implantation of the TVT based on responses to mailed questionnaires obtained at least 3 months after the operation. Overall, 26% of the patients reported improved intercourse, whereas worsening was only observed in one woman, and this was attributed to an increase in her incontinence [24]. Further evidence suggesting that the majority of women experience improvement in their sexual life after the TVT procedure has been reported by Glavind et al., who demonstrated that 78% of the patients scored higher in the please define questionnaire postoperatively. Similar to our results, partner-related domains were largely unchanged, which indicates that the incontinence does not have a significant negative effect on the partner, but primarily on the self-esteem of the affected women.

Some authors evaluated sensory changes in the vaginal and clitoral regions after implantation of the TVT as an important factor in the deterioration of sexual function. Bekker et al. investigated the neuroanatomy of the clitoris in relation to vaginal sling procedures. They postulate that the autonomic innervation of the vaginal wall may be disrupted by the TVT procedure, and this leads to an altered lubrication/swelling response [25]. Others observed decreased reactions to cold, warm, and vibratory stimuli in the clitoral region after TOT (transobturator tape) implantation. However, small sample size is an important limitation of the study [26]. Caruso et al. described a decrease in clitoral blood flow with lower mean pulsatility index, mean peak systolic velocity, and greater resistance index, observed six months after the TVT procedure when compared with the pretreatment values [27]. Our findings showed significant improvement of sexual life in the cured patients and no change in failed and complicated cases. On the other hand, high cure rate and low incidence of complications is the main limitation of the study. Still, our findings strongly suggest that successfully treated SUI results in sexual life-quality improvement. Therefore, we suggest that persistent incontinence might be considered as the main cause of lack of improvement in quality of sexual life, but analysis of a larger sample of failed cases is needed to prove this postulate. 

## 5. Conclusions

Our outcomes suggest that successful treatment of SUI with the midurethral sling statistically significantly improves the quality of sexual life. Treatment failure, resulting in persistent incontinence or complications such as OAB with urgency, appears to be aprobable cause of lack of improvement in the quality of sexual life. Further analysis of failed cases is needed to confirm these findings.

## Figures and Tables

**Table 1 jcm-09-01538-t001:** Demographic characteristics of the studied group.

	All (*n* = 171)	Sexually Active (*n* = 116)	Not Sexually Active (*n* = 55)	*p*
Age (years)	56.3 ± 11.2	52.7 ± 9.8	64.0 ± 10.1	<0.001
Height (cm)	163.4 ± 6.1	164 ± 6.1	162.1 ± 5.9	0.07 (ns)
Body weight (kg)	74.2 ± 12.7	72.7± 13.0	77.5 ± 1.7	<0.02
Body mass index (BMI)	27.8 ± 4.5	27.0 ± 4.5	29.5 ± 4.1	<0.001
No. deliveries (total)	2.1 ± 1.0	2.2± 1.0	2.0 ± 1.0	0.20 (ns)
No. of vaginal deliveries	1.9 ± 1.1	2.0 ± 1.1	1.9 ± 1.0	0.56 (ns)
No. of Cesarean sections	0.2 ± 0.5	0.2± 0.6	0.1± 0.3	0.15 (ns)
No. of instrumental deliveries (vacuum or forceps)	0.0 ± 0.1	0.0 ± 0.0	0.0 ± 0.1	0.15 (ns)
Average birth weight (g)	3362 ± 508	3367 ± 457	3351 ± 611	0.85 (ns)
Maximum birth weight (g)	3555 ± 528	3550 ± 467	3536 ± 648	0.76 (ns)

**Table 2 jcm-09-01538-t002:** The results of 1 h pad test and IIQ7 (Incontinence Impact Questionnaire-7) before and 6–12 months after TVT sling implantation.

	Before TVT			After TVT		
	All (*n* = 171)	Sexually active (*n* = 116)	Not sexually active (*n* = 55)	All (*n* = 133)	Sexually active (*n* = 86)	Not sexually active (*n* = 47)
Pad test (g)	63.3 ± 71.7	61.9 ± 69.0	66.5 ± 77.9	5.7 ± 33.8	3.2 ± 23.3	10.1 ± 46.8
Pad test ≤ 2 g				(121/133) 90.98%	(80/88) 92.0%	(41/45) 91.1%
IIQ7 score	71.4 ± 18.6	74.3 ± 32.9	79.0 ± 20.0	14.8 ± 22.4	13.8 ± 22.4	15.7 ± 23.3
Coital incontinence	96/171 (56.1%)			11/133 (8.6%)		

The differences are statistically significant (*p* < 0.001).

**Table 3 jcm-09-01538-t003:** The results of PISQ-IR (Pelvic Organ Prolapse/Urinary Incontinence Sexual Questionnaire, IUGA-Revised) before and 6–12 month after TVT implantation in the group of cured women.

Score	Before TVT	After TVT	*p*
Not sexually active *			
*n*	50	41	
Global quality	3.14 ± 0.98	2.58 ± 0.79	0.006
-Condition specific	2.07 ± 0.92	1.61 ± 0.88	0.006
-Condition impact	2.91 ± 1.09	1.44 ± 0.70	<0.001
-Partner related	2.72 ± 1.07	2.70 ± 1.06	0.87 (ns)
Not Sexually active * (patients with missing data in both pre- and post-operative surveys were excluded)			
*n*	39	39	
Global quality	3.06 ± 1.04	2.61 ± 0.78	0.019
-Condition specific	2.15 ± 0.98	1.68 ± 0.89	0.01
-Condition impact	3.01 ± 1.07	1.64 ± 0.83	<0.001
-Partner related	2.79 ± 1.02	2.77 ± 1.08	0.95
Sexually active **			
*n*	109	80	
Global quality	2.91 ± 0.80	3.31 ± 0.94	0.006
-Arousal, Orgasm	3.29 ± 0.80	3.60 ± 0.65	0.008
-Partner related	3.29 ± 0.58	3.43 ± 0.43	0.18 (ns)
-Condition specific	3.52 ± 1.09	4.54 ± 0.70	<0.001
-Condition impact	2.57 ± 0.77	3.42 ± 0.70	<0.001
-Desire	3.03 ± 0.62	3.07 ± 0.46	0.78 (ns)
Sexually active ** (patients with missing data in both pre- and post-operative surveys were excluded)			
*n*	77	77	
Global quality	2.94 ± 0.79	3.32 ± 0.93	0.015
-Arousal, Orgasm	3.20 ± 0.80	3.59 ± 0.65	<0.001
-Partner related	3.26 ± 0.52	3.41 ± 0.46	0.009
-Condition specific	3.56 ± 1.08	4.57 ± 0.69	<0.001
-Condition impact	2.59 ± 0.76	3.36 ± 0.76	<0.001
-Desire	2.97 ± 0.60	3.04 ± 0.45	0.287 (ns)

* Mean value ± standard deviation; higher scores indicate greater impact of condition on sexual function (High Scores Negative). ** Mean value ± standard deviation; higher scores indicate better sexual function (High Scores Positive).

**Table 4 jcm-09-01538-t004:** The results of PISQ-IR questionnaire before and 6–12 month after TVT implantation in the group of failed cases.

Score	Before TVT (*n* = 12)	After TVT (*n* = 12)	*p*
Not sexually active *			
*n*	4	4	
Global quality	2.27 ± 0.55	2.95 ± 0.90	0.48 (ns)
-Condition specific	2.30 ± 1.06	2.2 ± 0.67	0.61 (ns)
-Condition impact	2.67 ± 1.53	3.0 ± 0.96	0.48 (ns)
-Partner related	2.88 ± 0.85	3.0 ± 1.22	0.61 (ns)
Sexually active **			
*n*	8	8	*p*
Global quality	3.2 ± 0.40	2.94 ± 0.72	0.62 (ns)
-Arousal, Orgasm	2.53 ± 0.54	2.76 ± 0.25	0.07 (ns)
-Partner related	2.74 ± 0.62	2.92 ± 0.80	0.62 (ns)
-Condition specific	2.95 ± 1.42	3.8 ± 1.09	0.37 (ns)
-Condition impact	2.25 ± 0.69	2.46 ± 1.05	1.0 (ns)
-Desire	2.75 ± 0.36	2.68 ± 0.25	0.48 (ns)

* Mean value ± standard deviation, higher scores indicate greater impact of condition on sexual function (High Scores Negative). ** Mean value ± standard deviation, higher scores indicate better sexual function (High Scores Positive).

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
