# Peer review of "Impact of Midurethral Sling Implantation on Sexual Function in Women with Stress Urinary Incontinence"

_jcm, 2020, doi:10.3390/jcm9051538_

Round 1

Reviewer 1 Report

It is a very interesting study to evaluate the impact of MUS implantation on the sexual functions of women with SUI.

The structure and content of the article were properly described to understand the impact of MUS implantation on the sexual functions of women with SUI.

However, major revisions are necessary

1) Please describe the criteria of the sexually active group and non-active group

2) It is reasonable to analyze and describe the changes of variables before and after surgery in each patient who has completed f/u protocol after incontinence surgery.

3) Following the 2nd comment, Table 2 and 3 should be performed paired-T tests, and Table 4 should be performed Wilcoxon signed-rank tests.

4) The authors described that three of the 12 women who experienced surgical failure had urinary retention. Is that different from four cases who removed the tape due to urinary retention? Please clarify.

5) There are sentences in the Introduction and Discussion parts that require an additional reference. Please mark the reference carefully.

6) Pleased add the limitations of the current study.

Author Response

  1. Criteria of sexually active and non-active patients were added.
  2. According to the suggestion, we added the requested data to Table 3.
  3. Table 2. Differences  are statistically significant. We added the required information in table description.

    Table 3. We added the suggested data.  Paired T-Test was used for statistical analysis.

    Table 4. Wilcoxon signed-rank test had already been used for statistical analysis.

  4. In four cases we removed the sling during the hospital stay due to urinary retention and other symptoms bothering the patients (description added in the text). Those women had the sling located too close to the urethral lumen (< 3mm– determined in ultrasoud examination). In the 3 cases with urinary retention from the failed group the sling location was correct (>3mm from the urethral lumen). In these women, we had not observed retention nor other symptoms at discharge. Adequate inrofmation was added in the text.
  5. Additional refferences were added.
  6. Limitation of the study was added.

Reviewer 2 Report

This is a prospective cohort study to evaluate the influence of the successful TVT procedure on different aspects of sexual life and to analyse various aspects of sexual life in cases in whom failure of the anti-incontinence procedure was diagnosed.

The manuscript is overall well-written and flows well. It is meaningful data which will contribute to the growing body of literature concerning the relationship between midurethral sling implantation and sexual function in women. However, there are some issues with the reporting of the data. These are addressed in detail below.

Materials and Methods:

Line 67: How was the bladder filling of ca. 300 ml determined?

Line 87: Please elaborate on the surgical procedure, especially the 1/3 rule.

Line 90: ‘Follow-up visits were performed 6-12 months after surgery’. Please explain why such a wide interval was chosen as follow-up period.

Results:

Line 108: Please explain shortly the high loss to follow-up. This loss and its reasons should be elaborated on in the discussion paragraph.

Discussion:

The limitations of the study design are not discussed properly. It is advisable to incorporate this in the discussion, for completeness of the article.

Line 252-254: This postulation cannot be based on the results of the study. First of all, the number of patients with persistent incontinence is very small (line 147, 12 patients) causing the conclusions drawn to lack solidity. Secondly, this study has not specifically objectified the anatomically differences before and after sling surgery, therefore we believe this cannot be included.

Conclusion:

Line 257-259: This part of the conclusion isn’t solid and cannot be drawn from the results. The certainty with which the statement is made must be weakened.

Author Response

1. The method with which bladder filling was determined was described

2. The surgical procedure was described.

3. The follow-up visit was scheduled 9-12 months after the surgery. The mean follow-up was 10,4 months after the intervention. Few patient were seen earlier for their own convenience  (some of them live far form the clinic, some are not independent and need help to get to the clinic).

4. The loss to follow up was described. We excluded all the patient with incomplete questionairres (even with 1 or 2 missed answers).

5. Discussion: the limitation of the study was added.

6. The postulations were changed as recommended.

7. Conclusions were rewritten.

Round 2

Reviewer 1 Report

Thank you for your revision.
However, the current study still has major limitations in the methods.
The data contains patient cases that should not be included in the analysis.
After all, it is difficult to agree with the analyzed results.

Reviewer 2 Report

The authors addressed all the suggestions and the manuscript has been significantly improved.